# Damping Property of Cement Mortar Incorporating Damping Aggregate

**DOI:** 10.3390/ma13030792

**Published:** 2020-02-09

**Authors:** Yaogang Tian, Dong Lu, Jianwei Zhou, Yuxuan Yang, Zhenjun Wang

**Affiliations:** 1School of Materials Science and Engineering, Chang’an University, Xi’an 710064, China; dongluhit@163.com (D.L.); zjwang@chd.edu.cn (Z.W.); 2School of Civil Engineering, Harbin Institute of Technology, Harbin 150000, China; 3School of Materials Science and Engineering, Xi’an University of Architecture and Technology, Xi’an 710055, China; jianzhucat@163.com (J.Z.); yyx9202@163.com (Y.Y.)

**Keywords:** cement mortar, lightweight porous aggregate, damping aggregate, polymer emulsion, mechanical properties, damping property

## Abstract

This study proposes a new cement mortar incorporating damping aggregate (DA) and investigates the mechanical properties and damping property of the cement mortar. Four types of DA were prepared, lightweight aggregate presaturated water and three types of polymer emulsion. Further, the effects of polypropylene fiber and rubber powder on the performance of the cement mortar were studied. The experimental results showed that the damping ratio of specimens containing 70% DA was approximately three times higher than that of the reference mortar, with a slight decrease in the mechanical properties. Adding fiber was more effective than rubber powder in improving the damping ratio of the cement mortar, and the optimal dosage of fiber was 0.5%.

## 1. Introduction

Cement-based materials have been widely used in building construction and transport engineering due to their superior strength, anti-corrosion resistance, and low cost [1,2,3]. Nowadays, the characteristics of ultra-high rise, high-speed, and large-flow traffic in building construction and transport engineering became more prominent [4,5]. Therefore, cement-based materials and their structure inevitable exposure to dynamic loads during service [6,7,8]. The vibration energy induced by external dynamic loads cannot be dissipated effectively by cement-based materials itself on time. Thus, the original defects and micro-cracks of cement-based materials will spread rapidly and influence the durability and safety of the structure [7,9]. Thus, improving the damping property of cement-based materials is an urgent issue.

The damping property is the ability to translate the vibration energy of a system into other forms of energy [6,7,10], which effectively reduce the vibration of structure [6]. In view of the low damping capacity of cement-based materials [7], many studies reported that adding polymer emulsion [11] into cement-based materials is an effective way to improve its damping property due to the polymer emulsion has high flowability, excellent viscoelastic and damping performance, etc. [6,12,13,14,15]. For example, Tian et al. [6] prepared the lightweight aggregate presaturated polymer emulsion (LAP) and water (LAW) and investigated the replacement percentage of LAP on the damping ratio of concrete; the results indicated that the damping ratio of concrete increased with the increased of LAP%, and the increasing tendency produced by LAP% was much more obvious than that the LAW%, at the same volume fraction. Leiben et al. [14] investigated the effect of asphalt emulsion on the damping properties of cement asphalt emulsion mortars and found that the addition of asphalt emulsion can significantly improve the damping performance of cement mortar. Similar conclusions have been demonstrated by Fu et al. [16] and Hou et al. [13]. Further, the effect of fiber polymer on the damping property of cement-based materials has been reported by Jeon et al. [17]. However, the results indicated that the addition of a polymer emulsion could significantly reduce the mechanical properties of cement-based materials [10,18], which limits the applications of polymer emulsion in improving the damping property of cement-based materials.

Aggregate, as the principal raw material in cement-based materials, which accounts for about 60–70% of the total volume of cement-based materials [3,19]. Therefore, the type, shape, and size of aggregate significantly influence the performance of cement-based materials [20,21,22]. A lightweight aggregate has a low density and porous characteristics, which has been extensively used in building construction. Researches showed that the porous lightweight aggregate has high water-absorbing [6,23]. The presaturated water will release at a late stage of the hydration process of cement. Water-releasing role of lightweight aggregate provides a remarkable internal curing effect on cement-based materials during the hydration process of cement, which conducive to improve the microstructure and performance of cement-based materials [6]. Nowadays, the cement-based materials containing lightweight aggregate have been extensively studied for the physical and mechanical performance, as well as its durability [24,25]. However, studies on the effects of lightweight aggregate presaturated water and polymer emulsion on the damping property of cement mortar is still very limited, especially comparative performance of the cement mortar modified with polypropylene fiber and rubber powder.

In this study, for the first time, we prepared four types of damping aggregate (DA): lightweight aggregate presaturated water (WDA), lightweight aggregate presaturated styrene-acrylic emulsion (SEA), lightweight aggregate presaturated styrene-butadiene latex (SLA) and lightweight aggregate presaturated emulsified asphalt (EAA), and comparative performance (mechanical properties and damping property) of the cement mortar modified with polypropylene fiber and rubber powder. The experimental results showed that the potential of using DA to promote the damping property of cement-based materials. It is of great significance to further applications of cement-based materials with a high requirement of damping capacity.

## 2. Materials and Methods

### 2.1. Materials

The cement used in all mixes was ordinary Portland cement (P·O 42.5, Yaoxian cement plant, Tongchuan, China). Table 1 shows the chemical composition and the specific surface area of cement. The properties of cement meet the requirements of the Chinese National Standards GB175-2007 [26].

The water used was ordinary tap water. Three types of polymer emulsion were used to prepare DA, including styrene-acrylic ester polymer emulsion (SE), styrene-butadiene latex (SL), and emulsified asphalt (EA). The SE has an average particle diameter of 0.15 μm, solid content of 49.20%, and a pH of 7.50, respectively. The SL has an average relative molecular mass of 90,000 and a relative density of 0.93 g/m^3^, respectively. The EA has a 1.18 sieve test of 0.002, residue by distillation of 57.20% and an average particle diameter of 2.28 μm, respectively.

The fine aggregate used was natural sand (NS) and DA (lightweight aggregate presaturated water and polymer emulsion) with a fineness modulus of 2.65 and 2.68, respectively. Figure 1 presents the particle gradation of aggregate. Table 2 shows the properties of lightweight porous aggregates. Properties of the fine aggregates used complied with the Chinese National Standards GB/T 14684-2011 [27] and JGJ-2017 [28].

Polycarboxylate Superplasticizer (SP, Sobute New Materials, Nanjing, China) with water-reducing ratio of 28.0% was used. A polypropylene fiber (Dingqiang plant, Wuhan, China) with a density of 0.90 g/cm^3^ and a length of 6.0 mm was used to improve the damping property of cement mortar.

### 2.2. Preparation of DA

The preparation process of DA (Hongren Construction Engineering Co., Ltd., Xian, China), including lightweight aggregate presaturated water (WDA), lightweight aggregate presaturated styrene-acrylic emulsion (SEA, Melco Technology Co., Ltd., Wuhan, China), lightweight aggregate presaturated styrene-butadiene latex (SLA, Melco Technology Co., Ltd., Wuhan, China) and lightweight aggregate presaturated emulsified asphalt (EAA, Makepolo, Beijing, China), as described elsewhere [6]. Figure 2 shows the flow chart of the DA preparation. Figure 3 presents the three types of DA used in this study.

### 2.3. Mixture Preparation

Fifteen mixtures were considered to investigate the effects of fiber, rubber powder, and replacement percentage of DA on the mechanical properties and damping property of cement mortar. Table 3 presents the mixture proportions of the cement mortar.

### 2.4. Test Methods

#### 2.4.1. Mechanical Properties Test

Mechanical properties of the cement mortar were characterized by compressive strength and flexural strength in accordance with the Chinese National Standard JGJ/T70-2009 (ASTM C1761-2017) [29]. For compressive strength and flexural strength test, 40 mm × 40 mm × 40 mm cubic mold and 40 mm × 40 mm × 160 mm prism mold were used, respectively. The test specimens were demolded after 24 h of casting and then kept in a curing room (20 ± 1 °C, RH ≥ 95%) before testing.

#### 2.4.2. Damping Property Test

The damping property of the cement mortar was reflected by the damping ratio. A free-free beam vibration excited by an impact was conducted to investigate the damping ratio of the cement mortar (Figure 4). The dimension of the sample was 20 mm × 20 mm × 280 mm.

A small hammer was used to induce free vibration to the sample, and the acceleration sensor was attached to the sample by cyanoacrylate adhesive, the acceleration sensor in the center of the surface of the sample [6]. And the acceleration sensor had no influence on the free vibration of the beam.

To obtain the acceleration response signals as time-magnitude graphs during the experiment (Figure 5), Brüel & Kjær’s platform (PULSE Reflex Core, Bruel&Kjaer, Copenhagen, Denmark) for noise and vibration analysis was used [6]. The damping ratio of cement mortar was calculated by the half-power bandwidth method [30,31], as shown in Figure 6. It should be pointed out that the free vibration tests were repeated three times on each specimen, and the average was recorded from the successive results [9]. The damping ratio of the cement mortar was calculated as the following formula:(1)$$\xi =\frac{\left(f_{1}\;-\;f_{2}\right)}{2f_{0}}$$
where, ξ refers to the damping ratio, %; *f_1_*, and *f_2_* refer to the frequencies corresponding to an amplitude of *f_0_ /√2*, Hz; *f_0_* refers to the resonant frequency of the cement mortar, Hz.

## 3. Results and Discussion

### 3.1. Compressive Strength

Figure 7 shows the effects of aggregate type, replacement percentage of DA, fiber, and rubber content on the compressive strength of the cement mortar. Each compressive strength value corresponds to the average of three specimens.

To investigate the effect of aggregate type the compressive strength of the cement mortar, the replacement percentage of the aggregate was 30%, the content of the fiber and rubber was fixed at 0%. From Figure 7a, it could be observed that the compressive strength of the cement mortar containing 30% WDA slightly decreased, compared to the reference cement mortar after 28 days of curing. This could be attributed to the lightweight aggregate presaturated water released during the hydration process of the cement, increasing the effective water-cement ratio of the cement mortar. While the compressive strength of the specimens was approximately decreased by 15% (30% SLA) and 17% (30% EAA), respectively. This could be ascribed to the properties of the lightweight porous aggregate. As we all know, the interface transition zone (ITZ) between the cement paste and aggregate is the weakest region in the cement-based material. Therefore, introducing DA increases the proportion of weak interfaces. While this conducive to improve the damping property of cement-based materials, it will be discussed in Section 3.3. The compressive strength of the cement mortar modified with 30% SEA was approximately increased by 3%, the slight increment in compressive strength of the styrene-acrylic ester modified the cement mortar mainly due to the porosity reduction and the internal curing role [30,31].

To investigate the effect of replacement percentage of SEA on compressive strength of the cement mortar, the content of fiber and rubber powder was fixed at 0%. From Figure 7b, it can be seen that the compressive strength of cement mortar slightly increased by 1.1% (15% SEA) and 3.3% (30% SEA), respectively, compared with the reference cement mortar (0% SEA). While the compressive strength of specimens significantly decreased when the replacement percentage of SEA beyond 30%. These results have been reported by Tian et al. [6]. This mainly due to the residue polymer emulsion on the surface of porous lightweight aggregate increases as the replacement percentage of SEA increases, which reduces the bonding strength between the cement paste and aggregate. In addition, the lower elastic modulus of the DA (3–20 GPa) results in lower compressive strength of the cement mortar compared to the reference cement mortar.

To investigate the effect of polypropylene fiber on compressive strength of cement mortar modified with 30% SEA, the rubber powder was fixed at 0%, the fiber content of 0.25%, 0.5%, and 1% was used to mix the cement mortar. From Figure 7c, the compressive strength of the cement mortar modified with 30% SEA increased with the increase of the polypropylene fiber content. Compared with the cement mortar modified with 30% SEA, the compressive strength of the specimens with addition of 0.25%, 0.5%, and 1% fiber increased by 0.4%, 5.5%, and 8.7%, respectively, after 28 days curing. This is mainly due to the polypropylene fiber plays a “bridge joint” role in the cement mortar [17], which can connect dispersive SEA to exert the function of synergies. It can also effectively prevent the extension of cracks in the cement mortar and enhance the compressive strength of the cement mortar.

To investigate the effect of rubber powder on compressive strength of cement mortar modified with 30% SEA, the content of fiber was fixed at 0%, the rubber powder content of 2.5%, 5.0%, and 7.5% was used to mix the cement mortar. From Figure 7d, it can be seen that the compressive strength of the sample with the addition of 2.5% rubber powder sharply decreased. That is to say, introducing rubber powder had a significant negative influence on the compressive strength of the cement mortar. This could be ascribed to the weak adhesion between the rubber powder and cement paste [20]. The initial cracks will rapidly spread around the rubber powder particles when the cement mortar is suffering from external loads, which lead to a reduction in the compressive strength.

### 3.2. Flexural Strength

Figure 8 shows the effects of aggregate type, replacement percentage of DA, fiber, and rubber content on the flexural strength of cement mortar. Each flexural strength value corresponds to the average of three specimens.

Compared with the reference cement mortar after 28 days of curing, flexural strength of cement mortar containing DA slightly decreased by 3.5% (30% WDA), 3.5% (30% SLA), and 5.3% (30% EAA). While the cement mortar modified with 30% SEA, its flexural strength slightly increased by 1.7%. A significant drop in flexural strength of the cement mortar could be obtained when the replacement percentage of SEA beyond 30%, flexural strength of the specimens containing 50%, 70% and 100% SEA decreased by 14.0%, 24.6%, and 29.8%, respectively, after 28 days of curing, compared with the specimen containing 0% SEA. The effects of fiber content and rubber powder content on the flexural strength of the cement mortar modified with 30% SEA were similar to that of the compressive strength, indicating that the addition of fiber was more effective than rubber powder in improving the mechanical properties of the cement mortar.

### 3.3. Damping Ratio

Figure 9 shows the effects of aggregate type, replacement percentage of DA, fiber, and rubber content on the damping ratio of the cement mortar. Each damping ratio value corresponds to the average of three specimens.

To investigate the effect of the aggregate type on the damping ratio of the cement mortar, the replacement percentage of the aggregate was 30%, the content of the fiber and rubber were fixed at 0%. Compared to the reference cement mortar, the damping ratio of the specimens increased by 8.0% (30% WDA), 40.0% (30% SEA), 20.1% (30% SLA), and 28.3% (30% EAA), respectively, after seven days of curing, while sharply increased by 31.6% (30% WDA), 52.6% (30% SEA), 34.2% (30% SLA) and 57.9% (30% EAA) in each case after 28 days of curing, as shown in Figure 9a. This could be attributed to the polymer emulsion presaturated by the lightweight aggregate, as shown in Figure 10. The energy dissipation is from the zigzag molecule chains of polymer emulsion, producing stretching, twisting, and other deformation, as well as the relative slip and torsion among the molecular chains [11,12].

Figure 9b shows the effect of replacement percentage of DA on the damping ratio of cement mortar. It can be observed that the damping ratio of the cement mortar modified with DA distinctly higher than that of the reference cement mortar. The damping ratio of the cement mortar increased with the increased of replacement percentage of DA. The damping ratio of the cement mortar approximately increased by 30% when the replacement percentage of the DA was below 30%, while it significantly increased when the replacement percentage of the DA was beyond 30%. The damping ratio of the specimens increased by 94.7%, 176.3%, and 234.2% for the specimens containing 50% SEA, 70% SEA, and 100% SEA, respectively, compared with the reference cement mortar after 28 days of curing.

Figure 9c shows the effect of polypropylene fiber content on the damping ratio of cement mortar modified with 30% SEA. It can be observed that the damping ratio of the cement mortar indicated an increasing trend as the polypropylene fiber content increased. The increasing tendency was very slight when the fiber content less than 0.5%, while a sharp increment could be found at the specimen with the addition of 1% polypropylene fiber. It can be concluded that the polypropylene fiber is beneficial to increase the damping ratio of the cement mortar.

Figure 9d shows the effect of rubber powder content on the damping ratio of cement mortar with 30% SEA. It can be seen that the damping ratio of cement mortar significantly increased by 54.2% (2.5% rubber powder), 74.6% (5.0% rubber powder), and 115.3% (7.5% rubber powder), respectively, compared with the cement mortar containing 0% rubber powder. It can be concluded that the addition of rubber powder is beneficial to increase the damping ratio of the cement mortar. Rubber powder, as a kind of macromolecule material, has a low elasticity modulus and the performance of scalability, making its own a good damping capacity [12]. Further, the big gap in the modulus of elasticity between the rubber powder and the cement paste results in the relative deformation of the cement mortar. These all significantly improve the energy dissipation of the cement mortar.

## 4. Damping Mechanism

The cement mortar modified with DA is comprised of five-phase mediums, such as DA, NS, cement paste and ITZs (ITZ between the cement paste and NS, ITZ between the cement paste and DA). The big gap of the elastic modulus of the five-phase mediums, for example, the elastic modulus of DA (3–20 GPa) is far smaller than that of NS (40–100 GPa), will be inhomogeneous in the cement mortar, and would increase the damping property of the sample due to the relative deformation between different phases [6].

The DA was prepared by a lightweight porous aggregate and polymer emulsion. It should be noted that the lightweight aggregate exists a large number of openings pores and grooves, as seen in Figure 10. The hydration products of cement can penetrate into the opening pores of the lightweight aggregate [9,15,21,32], which can increase the ITZ area of the cement mortar. Therefore, the addition of DA increases the damping property by the ITZ friction of the specimen. In addition, a large number of calcium hydroxide (CH) crystals and a large number of micro-cracks and voids in the ITZ [9,33]. This was conducive to improving the energy dissipation of cement-based materials during vibration.

The DA presaturated water or polymer emulsion, introducing water to cement mortar. Previous reports showed that the addition of water can improve the damping property of cement-based materials [6]. As we all know, a polymer emulsion has a super flowability and high damping characteristic. When the lightweight aggregate presaturated with polymer emulsion and is introduced into the cement mortar, it will form a polymer film in the pores of the surface of aggregate, which attach to one side of the lightweight aggregate, just like the free-layer or extension of the damping structure, as shown in Figure 11. This can alter, limit, or disperse the vibration energy of the structure [18]. The hydration products of cement can penetrate into the DA throughout the opening pores. At the same time, some of the polymer emulsion adhere to the lightweight porous aggregate, and then a mesh-like structure with cement paste and DA is formed. Namely, a viscoelastically constraint-layer (sandwich) damping structure in the cement mortar is formed [6], as shown in Figure 11, which can dissipate vibration energy through the viscoelastic polymer film adhered in the pore of the DA.

## 5. Conclusions

This paper presented a new cement mortar incorporating DA with a high damping ratio, further, comparative performance of cement mortar modified with fiber and rubber powder. A series of experimental tests of the compressive strength, flexural strength, and the damping ratio were carried out. Based on experimental tests and damping mechanism analyses, the following conclusions can be drawn:(1)The compressive strength of specimens containing 30% DA all retained more than 35 MPa. The addition of fiber had a marginal effect on the compressive strength of the cement mortar, while the flexural strength of specimens approximately increased by 25% when the fiber content beyond 0.5%. Adding rubber powder had a negative effect on the compressive and the flexural strength of the cement mortar, especially when the rubber powder content beyond 2.5%.(2)The damping ratio of the specimen showed an increasing trend as the percentage of DA increased. The increasing tendency was slight when the percentage of DA below 30%, while a sharp increase of damping ratio could be found at the specimen containing 50% DA. Damping ratio of the specimen containing 70% SEA was approximately three times higher than that of reference mortar. Indicating that the cement mortar incorporating DA can be used to improve the damping property of structures.(3)The new cement mortar containing the DA developed had a superior damping property compared to that of the normal cement mortar. The addition of fiber or rubber powder can further improve the damping property of the cement mortar. Adding fiber was more effective than rubber powder in improving the damping ratio of the cement mortar, and the optimal dosage of the fiber was 0.5%. The damping ratio of the specimen containing 1.0% fiber reached to 10.7% after 28 days of curing, which was approximately one times higher than that of the cement mortar with 30% SEA, with a slight influence in the mechanical properties.

## Figures and Tables

**Figure 1 materials-13-00792-f001:**
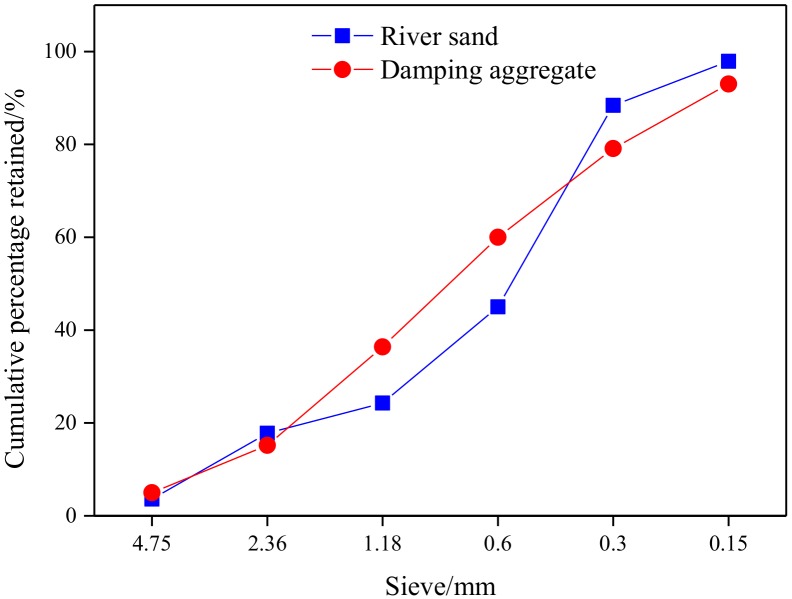
Particle gradation of aggregate.

**Figure 2 materials-13-00792-f002:**
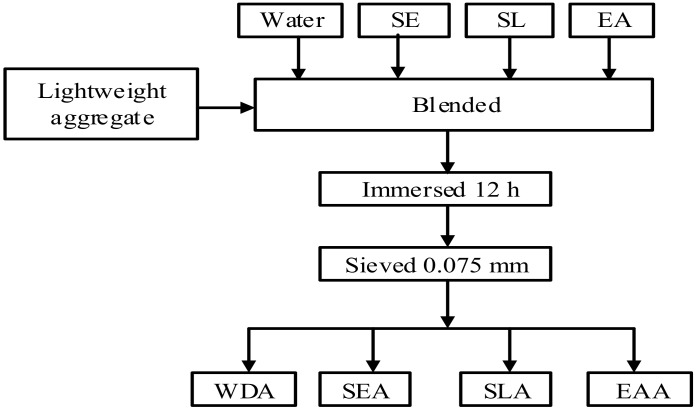
Flow chart of damping aggregate preparation.

**Figure 3 materials-13-00792-f003:**
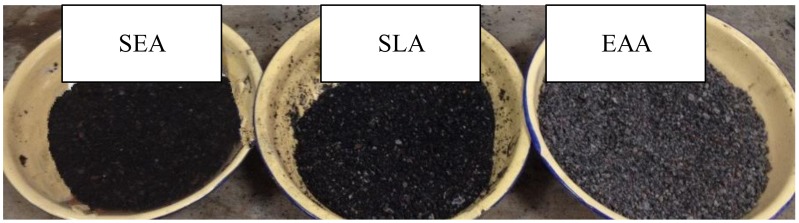
Damping aggregate used in this study.

**Figure 4 materials-13-00792-f004:**
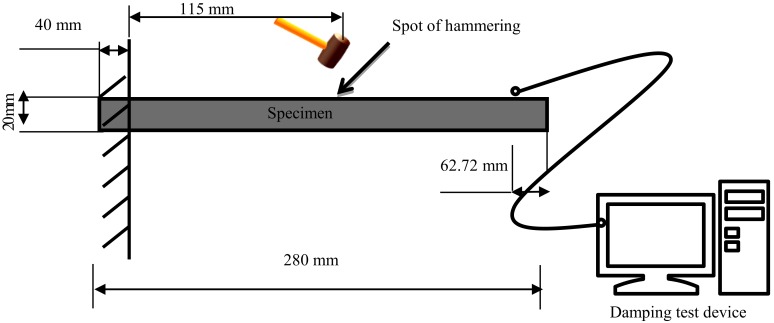
Schematic of the experimental setting and specimen dimension for the free vibration test.

**Figure 5 materials-13-00792-f005:**
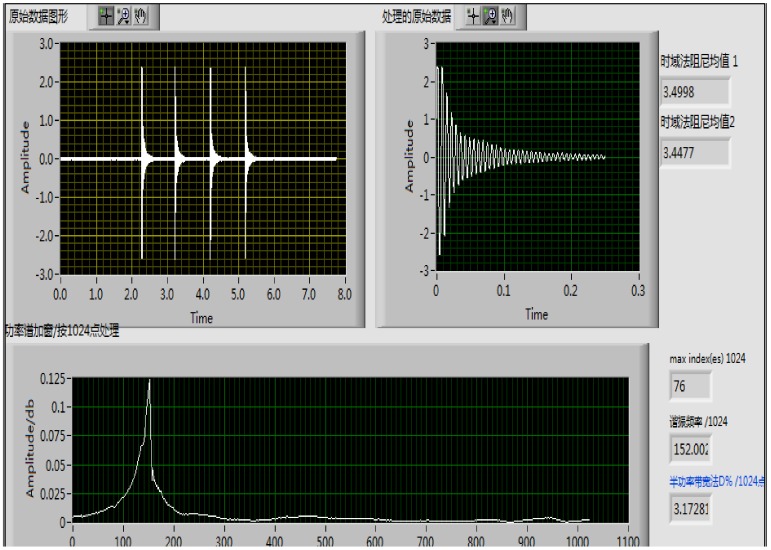
Acceleration response signals as time-magnitude.

**Figure 6 materials-13-00792-f006:**
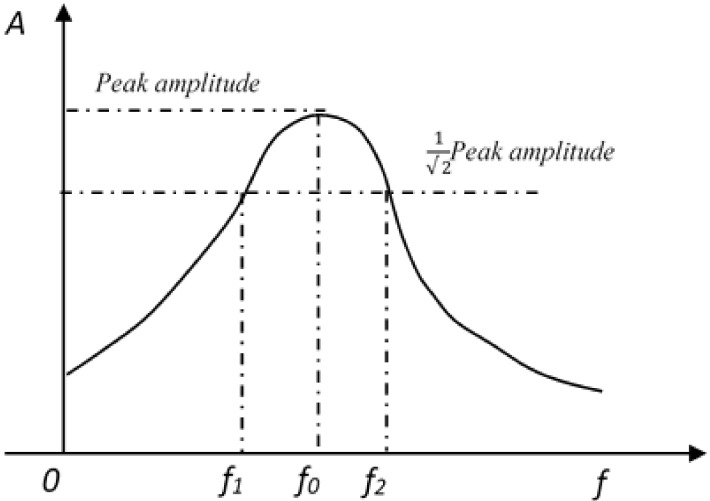
The half-power bandwidth method for the estimation of the damping ratio of the cement mortar.

**Figure 7 materials-13-00792-f007:**
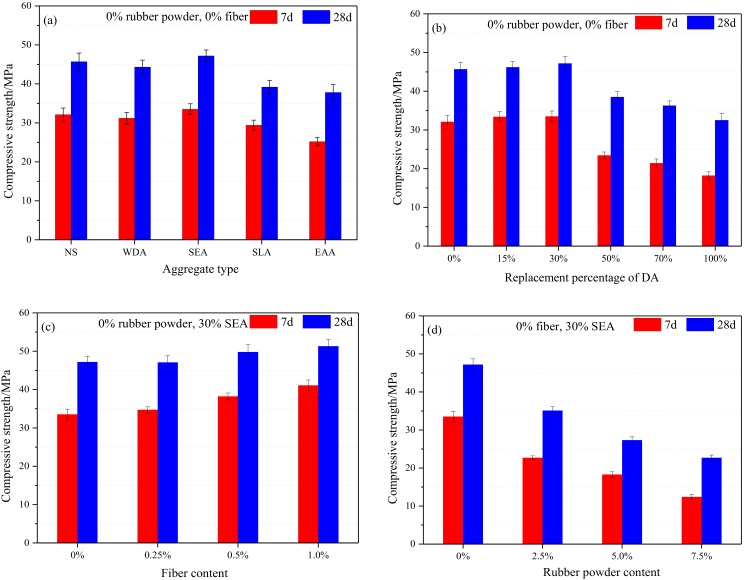
Compressive strength of cement mortar: (**a**) aggregate type; (**b**) replacement percentage of DA; (**c**) fiber content, and (**d**) rubber powder content.

**Figure 8 materials-13-00792-f008:**
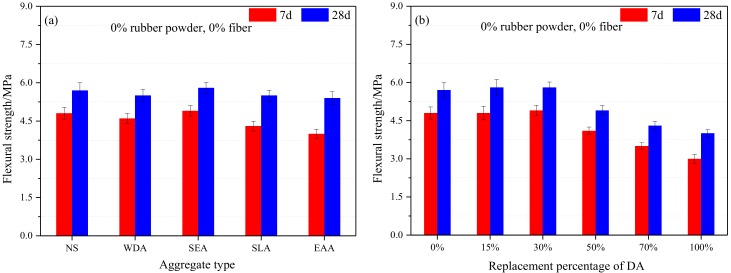
Flexural strength of the cement mortar: (**a**) aggregate type; (**b**) replacement percentage of DA; (**c**) fiber content, and (**d**) rubber powder content.

**Figure 9 materials-13-00792-f009:**
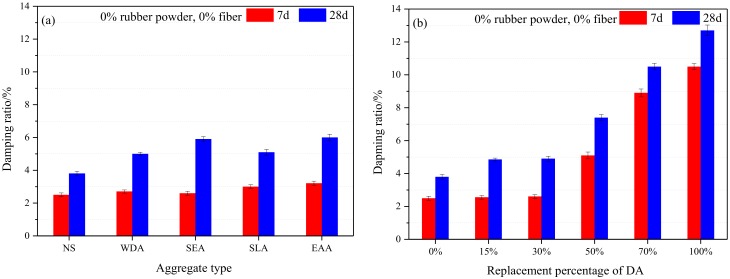
Damping ratio of the cement mortar: (**a**) aggregate type; (**b**) replacement percentage of DA; (**c**) fiber content, and (**d**) rubber powder content.

**Figure 10 materials-13-00792-f010:**
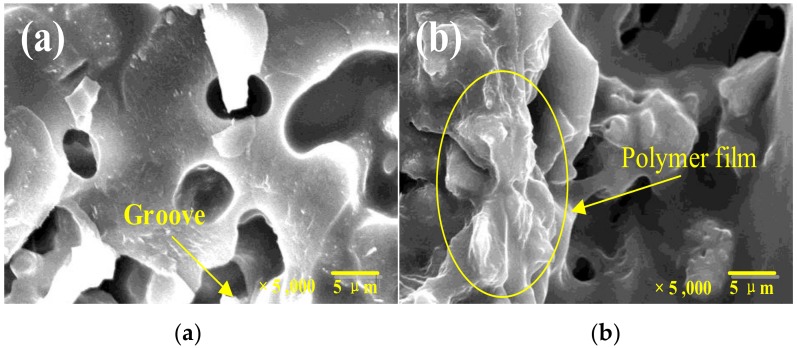
Secondary Electron images of damping aggregate: (**a**) lightweight aggregate and (**b**) lightweight aggregate presaturated with polymer.

**Figure 11 materials-13-00792-f011:**
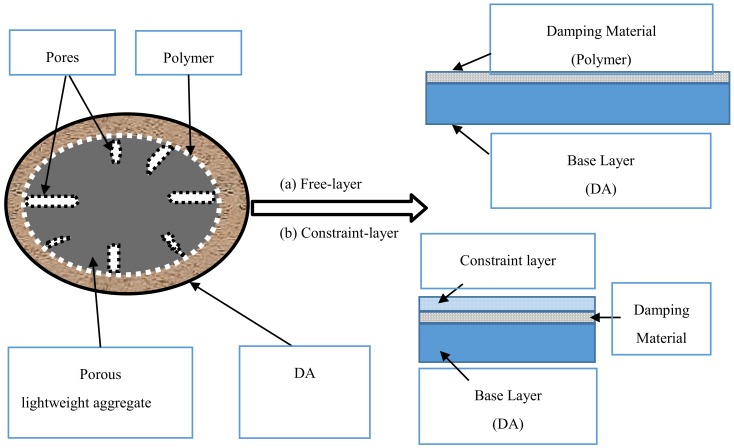
Damping structure among the DA, polymer emulsion, and hydration products of the cement in mortar.

**Table 1 materials-13-00792-t001:** Chemical composition and specific-surface area of cement.

Mass (wt.%)							Specific Surface Area (m^2^/kg)
SiO_2_	Al_2_O_3_	Fe_2_O_3_	CaO	MgO	SO_3_	Loss	
21.38	5.74	4.01	59.79	3.47	2.11	2.68	360.00

**Table 2 materials-13-00792-t002:** Properties of the lightweight porous aggregate.

Crushing Strength (MPa)	Particle Density (g/cm^3^)	Bulk Density (g/cm^3^)	Water Absorption (%)			Polymer Emulsion Absorption (%)		
			30 min	1 h	24 h	30 min	1 h	24 h
2.3	1.2	0.5	4.3	5.3	5.8	0.9	1.5	2.0

**Table 3 materials-13-00792-t003:** Mixture proportions of cement mortar (kg/m^3)^.

Mix Code	Aggregate	NS	DA	Water	Cement	SP (%)	Fiber (%)	Rubber Powder (%)
1	NS	1350.0	0	149.1	450	0.5	-	-
2	SEA	1147.5	202.5	149.1	450	0.5	-	-
3	SEA	945.0	405.0	149.1	450	0.5	-	-
4	SEA	675.0	675.0	149.1	450	0.5	-	-
5	SEA	405.0	945.0	149.1	450	0.5	-	-
6	SEA	0	1350.0	149.1	450	0.5	-	-
7	SEA	945.0	187.4	149.1	450	0.5	0.25	-
8	SEA	945.0	187.4	149.1	450	0.5	0.50	-
9	SEA	945.0	187.4	149.1	450	0.5	1.00	-
10	SEA	945.0	187.4	149.1	450	0.5	-	2.5
11	SEA	945.0	187.4	149.1	450	0.5	-	5.0
12	SEA	945.0	187.4	149.1	450	0.5	-	7.5
13	WDA	945.0	187.4	149.1	450	0.5	-	-
14	SLA	945.0	187.4	149.1	450	0.5	-	-
15	EAA	945.0	187.4	149.1	450	0.5	-	-

Note: All aggregates used were prepared to saturate surface dry conditions before mixing, the NS was replaced by WDA, SEA, SLA and EAA, respectively (wt.%).

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
