# Peer review of "Damping Property of Cement Mortar Incorporating Damping Aggregate"

_materials, 2020, doi:10.3390/ma13030792_

Round 1

Reviewer 1 Report

I would recommend the authors to relate their results to those of other researchers in discussion section more detailed.

Author Response

Thank you for your comments and suggestions, please see the attachment.

Reviewer 2 Report

The paper present an experimental investigation aimed at study the mechanical performances (compressive and flexural strength and damping ratio) of several mixes containing lightweight aggregated presaturated with polymer emulsion, fibers and rubber powder.

The study presents interesting results but must be strongly revised.

There are several typos, grammar mistakes and even a sentence repeated (see pag. 5).

The following points must be clarified:

-§2.4 International readers are not familiar with Chines standard, I suggest to the author to add the size of the specimens for compressive and flexural strength

- Fig. 7, 8 and 9 should be revised. – All the pictures should have the same range, in addition on picture a label with the “fixed parameters” should be added.

For instance in Fig. 7c a label with “30% SEA” should be added. I suppose that in that case the rubber powder content was 0%. This information must be added.

A more deep discussion of the results, in the particular their potential application in practice must be discussed.

Line 197 – replace compressive strength with damping ratio

Author Response

(The authors gave the same response as above.)

Reviewer 3 Report

This article is well written but I would recommend using a smaller number of abbreviations. The use of many abbreviations makes the article somewhat confusing. I would recommend at least all the shortcuts redefined in the section Results and discussion for better clarity.

Line 31 contains twice the word than

In the method section it is necessary to better define the damping test. It is not clear from the whole article whether the increase in damping ratio leads to improvement or deterioration of property. This would be good to write more about the applicability of these tests and the information resulting from these results.

Also results are not so clearly presented because there is not clear of if there has been an improvement of properties.

It would be good to add more information about influence of aggregates and polymers on teh mechanical strength. I think the explanation in the article is not sufficient.

Author Response

Thank you for your comments and suggestions,please see the attachment.

Reviewer 4 Report

The English needs to be profoundly revised. 

Author Response

(The authors gave the same response as above.)

Reviewer 5 Report

The manuscript needs extremaly extensive English editing, including the title.

Abstract and Conclusion should be shorter and without too much values, present just general findings and trends.

Author Response

(The authors gave the same response as above.)

Round 2

Reviewer 2 Report

The paper has been improved.

I suggest to the authors to check again English.

Author Response

We very much appreciate the positive comments,as well as the  valuable suggestions, which has been very helpful in improving the quality of this  paper.

We carefully checked and corrected the English throughout the paper.

We hope you will be satisfied with the revisions for the original manuscript.

Thanks and best regards,

Your  Sincerely,

Dong Lu

2020.26

Reviewer 5 Report

Authors improved the manuscript, but another round of English editing is needed - e.g. power instead of powder.

Author Response

We are so deeply apppreciate the time and effort you have spent in reviewing our manuscript. Thank you very much for your comments and suggestions, all the comments and suggestions are very important, which has been very helpful in improving the quality of this paper.

Rubber powder is a proper noun, as seen in the following paper. The "powder" instead of "power" throughout the paper, except for the "half-power  bandwidth method" used in this paper.

1. X.L., Liu X.P., Wang. Reclcling of waste rubber powder by mechano-chemical modification.  journal of cleaner production, 245, 2020, 118716. 

2. T, Gupta, S, Siddique. Behavior of waste rubber powder and hybrid rubber concrete in aggressive environment. construction and building materials, 217, 2019, 283-291.

We hope the editor and reviewer will be satisfied with the  revisions for the original manuscript.

Thanks and best regards,

Yours  Sincerely,

Dong Lu

2020.2.6